# Antioxidant Activity of an Aqueous Leaf Extract from *Uncaria tomentosa* and Its Major Alkaloids Mitraphylline and Isomitraphylline in *Caenorhabditis elegans*

**DOI:** 10.3390/molecules24183299

**Published:** 2019-09-10

**Authors:** Bruna C. Azevedo, Mariana Roxo, Marcos C. Borges, Herbenya Peixoto, Eduardo J. Crevelin, Bianca W. Bertoni, Silvia H. T. Contini, Adriana A. Lopes, Suzelei C. França, Ana M. S. Pereira, Michael Wink

**Affiliations:** 1Departamento de Biotecnologia em Plantas Medicinais, Universidade de Ribeirão Preto, Ribeirão Preto 14096-900, São Paulo, Brazil; 2Faculdade de Medicina de Ribeirão Preto, Universidade de São Paulo, Ribeirão Preto 14049-900, São Paulo, Brazil; 3Institute of Pharmacy and Molecular Biotechnology, Heidelberg University, D-69120 Heidelberg, Germany; 4Faculdade de Filosofia, Ciências e Letras de Ribeirão Preto, Departamento de Química, Universidade de São Paulo, Ribeirão Preto 14049-900, São Paulo, Brazil

**Keywords:** *Uncaria tomentosa*, *Caenorhabditis elegans*, antioxidants, oxidative stress, mitraphylline, isomitraphylline

## Abstract

*Uncaria tomentosa* (Rubiaceae) has a recognized therapeutic potential against various diseases associated with oxidative stress. The aim of this research was to evaluate the antioxidant potential of an aqueous leaf extract (ALE) from *U. tomentosa*, and its major alkaloids mitraphylline and isomitraphylline. The antioxidant activity of ALE was investigated in vitro using standard assays (DPPH, ABTS and  FRAP), while the in vivo activity and mode of action were studied using *Caenorhabditis elegans* as a model organism. The purified alkaloids did not exhibit antioxidant effects in vivo. ALE reduced the accumulation of reactive oxygen species (ROS) in wild-type worms, and was able to rescue the worms from a lethal dose of the pro-oxidant juglone. The ALE treatment led to a decreased expression of the oxidative stress response related genes *sod-3*, *gst-4,* and *hsp-16.2*. The treatment of mutant worms lacking the DAF-16 transcription factor with ALE resulted in a significant reduction of ROS levels. Contrarily, the extract had a pro-oxidant effect in the worms lacking the SKN-1 transcription factor. Our results suggest that the antioxidant activity of ALE in *C. elegans* is independent of its alkaloid content, and that SKN-1 is required for ALE-mediated stress resistance.

## 1. Introduction

*Uncaria tomentosa* (Willd.) DC. (Rubiaceae), commonly known as cat´s claw, is widely distributed in Latin America, especially in the Brazilian Amazon [1]. The species is mainly rich in indole and oxindole alkaloids, but also contains triterpenoids derived from quinovic acid and polyphenols, particularly tannins [2,3,4,5]. Ethnopharmacological surveys have shown that indigenous populations of Amazonia use *U. tomentosa* to treat asthma, arthritis, dermatitis, inflammation of the urinary tract, and benign and malignant tumors [6]. In addition, pharmacological studies on different *U. tomentosa* extracts have confirmed their anti-asthmatic, anti-diabetic, anti-microbial, anti-inflammatory, anti-cancer, antioxidant, and immunostimulant properties, as well as with neuroprotective effects against Parkinson’s and Alzheimer’s disease [7,8,9].

The radical scavenging activity of *U. tomentosa* extracts was previously characterized in vitro through 1,1-diphenyl-2-picrylohydrazyl (DPPH), superoxide dismutase (SOD), and 2,2′-azinobis-(3-ethylbenzothiazoline-6-sulfonic acid) (ABTS) assays [10,11,12,13,14,15]. Using cell culture models, Bors et al. [16] confirmed the antioxidant and protective effects of ethanol and aqueous preparations from the leaves and stems of *U. tomentosa* on 2,4-dichlorophenol-stressed human erythrocytes, by demonstrating a reduction of the reactive oxygen species (ROS) levels and hemolysis upon treatment with the extracts. Furthermore, Dreifuss et al. [17] reported on their antineoplastic activity against Walker-256 cancer cells and on the free-radical (DPPH) scavenging properties of a hydroalcoholic extract from *U. tomentosa* bark. The antioxidant and anti-genotoxic effects of an analogous extract also protected zebrafish (*Danio rerio*) against the oxidative damage induced by the glyphosate-based herbicide Roundup^®^, by impeding the reduction of total thiols in the brain, the glutathione peroxidase activity in the liver, and the increase in lipid peroxidation in both the brain and liver [18].

Although many studies have reported the antioxidant potential of *U. tomentosa* extracts, the molecular mechanisms underlying these activities have remain poorly understood.

*Caenorhabditis elegans* is now widely used and recognized as a model organism in various areas of pharmacological research [19,20]. It has been successfully applied to investigate the antioxidant and anti-aging activities of plant secondary metabolites as an alternative to higher eukaryotic model organisms [21,22].

In this context, we assume that *C. elegans* could be a suitable model system to elucidate the molecular mechanisms of *U. tomentosa* against oxidative stress. In order to test this hypothesis, we evaluated the effect of *U. tomentosa* extract and its major alkaloids, mitraphylline and isomitraphylline, on the intracellular levels of the ROS and survival rate upon a lethal dose of the pro-oxidant juglone in wild-type worms (N2). To further investigate the mechanisms of action behind the stress resistance, the expression of the stress-responsive genes superoxide dismutase-3 (*sod-3*), glutathione S-transferase (*gst-4*), and heat-shock protein-16.2 (*hsp-16.2*) was evaluated in transgenic worm strains (CF1553, CL2166, and TJ375) carrying a green fluorescent protein (GFP) reporter. Moreover, the involvement of DAF-16/FOXO and SKN-1/Nrf2 transcription factors was studied, using loss-of-function mutants (CF1038 and EU1).

## 2. Results

### 2.1. Chemical Characterization of a Leaf Extract from U. tomentosa

Ultra-performance liquid chromatography-mass spectrometry (UPLC-MS) allowed for the identification of mitraphylline (MIT), isomitraphylline (ISOMIT), and isorhynchophylline (Figure 1A–C) in the extract (Figure 1D). The oxindole alkaloid isorhynchophylline used in this study was isolated from the ALE extract.

### 2.2. In Vitro Antioxidant Activity

The in vitro antioxidant capacity of ALE was evaluated by DPPH, ABTS, and ferric reducing antioxidant power (FRAP) assays. The extract showed a moderate antioxidant activity when compared to ascorbic acid, a standard antioxidant used as a positive control (Table 1). A high phenolic content of 153.51 µg gallic acid equivalents (GAE)/mg extract was observed using the Folin–Ciocalteu method.

### 2.3. In Vivo Antioxidant Activity of ALE, Mitraphylline, and Isomitraphylline in C. elegans

#### 2.3.1. Intracellular Accumulation of ROS in Wild-Type Worms

The ROS levels of the N2 worms that were pre-treated with ALE, and the alkaloids were measured under physiological conditions using the cell-permeable reagent 2′,7′-dichlorofluorescein diacetate (H2DCF-DA). After entering the cell, H2DCF-DA was converted by cellular esterase to an intermediate non-fluorescent form that was later oxidized by ROS to a fluorescent compound, DCF. The fluorescent intensity of this oxidative derivative is correlated with the intracellular amount of ROS of a living organism. Treatment with 40 μg/mL of ALE significantly reduced the intracellular accumulation of ROS by 25.58% ± 2.16, respectively, in comparison with the untreated control (*p* < 0.0001; Figure 2A). In contrast, its major alkaloids, mitraphylline and isomitraphylline, failed to reduce the ROS levels (Figure 2B). Rather, the treatment with 10 μg/mL isomitraphylline displayed a pro-oxidant effect, increasing the ROS levels by 22.6% ± 4.13, in comparison with the DMSO control (*p* < 0.0001). Epigallocatechin gallate (EGCG), a polyphenol extracted from green tea, purchased from Sigma-Aldrich (St. Louis, MO, USA; CAS number 989-51-5; 95.0%purity), was used as a positive control. Compared with the untreated control, the EGCG treatment at 50 μg/mL reduced the ROS levels by 43.36% ± 2.16.

#### 2.3.2. Survival Rate of Wild-Type Worms Exposed to a Lethal Dose of Juglone

In order to evaluate the potential of ALE and the alkaloids to rescue the N2 worms from a lethal dose of juglone (80 μmol), the survival rate of the worms pre-treated with the extracts was evaluated after 24 h of juglone-induced oxidative stress. The survival rate of the juglone-stressed worms that were pre-treated with 80 μg/mL ALE was 53.91% ± 5.63. This value was comparable with the survival rates of the worms pre-treated with 50 μg/mL EGCG (58.84% ± 6.34), and was significantly higher (*p* < 0.0001) than those of the juglone-treated worms (30.59% ± 1.75; Figure 3A). Mitraphylline and isomitraphylline at 1 and 2 μg/mL were unable to rescue the worms (Figure 3B).

### 2.4. Mechanism of the Antioxidant Activity of ALE in C. elegans

#### 2.4.1. Effect of ALE on the Expression Levels of the Oxidative Stress Resistance Related Genes: *sod-3, gst-4*, and *hsp-16.2*

The ability of ALE to modulate the expression of oxidative stress resistance related genes was investigated using the transgenic strains CF1553, CL2166, and TJ375 carrying a GFP reporter fused with the promoter regions of *sod-3*, *gst-4,* and *hsp-16.2*, respectively.

Under basal stress conditions, the CF1553 (*sod-3p*::GFP) worms pre-treated with 40 and 80 μg/mL of ALE showed a minor but significant reduction of GFP intensity, when compared with the untreated control (Figure 4A). At 80 μg/mL of ALE, the expression of *sod-3* (*p* < 0.001) was reduced by 12.08% ± 2.20. In comparison with the untreated control, EGCG at 50 μg/mL reduced the fluorescence intensity by 23.82% ± 2.56 (*p* < 0.0001). Exposure to a mild dose of juglone (20 μM) increased the *sod-3* expression levels by 45.82% ± 1.58, in comparison with the untreated control (*p* < 0.0001; Figure 4B). Under juglone-induced oxidative stress, treatment with 80 μg/mL of ALE showed no effect on the expression of *sod-3*.

In *C. elegans*, *hsp-16.2* and *gst-4* are expressed in response to several types of environmental stress, including oxidative stress. Therefore, a mild dose of the pro-oxidant juglone was used to trigger the expression of these genes. The pre-treatment of juglone-stressed CL2166 worms (*gst-4p*::GFP) with ALE in the concentration range of 40 to 100 μg/mL induced significant reductions (*p* < 0.0001) of 32.57% to 63.31% in the intensity of the GFP fluorescence in comparison with the juglone- treated worms (Figure 4C). The EGCG treatment at 50 μg/mL also reduced the GFP fluorescence by 81.01% in comparison with the juglone-treated worms. In the juglone-stressed TJ375 worms (*hsp-16.2p*::GFP) pre-treated with 40 and 80 μg/mL of ALE, the intensity of the GFP fluorescence showed significant (*p* < 0.0001) reductions of 41.77% ± 3.02 and 41.35% ± 2.72, respectively, in comparison with the juglone-treated worms (Figure 4D). At the highest concentration tested, ALE revealed no effect on the *hsp-16.2* expression levels. Pre-treatment with 50 μg/mL EGCG reduced the GFP fluorescence by 35.02% ± 3.75.

#### 2.4.2. Involvement of DAF-16 and SKN-1 in the Antioxidant Activity of ALE

DAF-16/FOXO and SKN-1/Nrf2 are key transcription factors in the modulation of oxidative stress resistance and longevity in *C. elegans*. To investigate whether stress resistance mediated by ALE depends on DAF-16 and/or SKN-1, mutant strains lacking *daf-16* (CF1038) and *skn-*1 (EU1) genes were used to quantify the intracellular ROS levels and the survival rate of juglone-stressed worms.

In CF1038 worms, the ROS levels were significantly lower for all of the ALE concentrations tested, in comparison with the untreated control (Figure 5A). At 100 μg/mL, the highest reduction in fluorescence intensity was observed (27.06% ± 2.21, *p* < 0.0001). On the other hand, in the EU1 worms, the ALE at 40 μg/mL showed a pro-oxidant activity, with the ROS accumulation increasing by 15.2% ± 4.03 in comparison with the untreated control (*p* < 0.01; Figure 5B).

The survival rates of the CF1038 worms, pre-treated with ALE, were slightly higher than those of the untreated worms, however not in a significant way (Figure 6A). In turn, the EU1 worms treated with 100 μg/mL of ALE showed significantly higher survival rates in comparison with the juglone-treated worms (*p* < 0.05), but were substantially lower than those achieved by EGCG pre-treatment (80.98% ± 3.84; Figure 6B).

### 2.5. Effect of ALE on C. elegans Body Size and Escherichia coli OP50 Growth

The ALE treatment did not reduce the body size of the N2 worms pre-treated with ALE. In fact, at 40 μg/mL of ALE, the size of the worms significantly increased in comparison with the untreated worms (*p* < 0.0001; Figure 7). ALE did not exhibit an antibacterial activity against *Escherichia coli* OP50.

## 3. Discussion

Most studies on the *U. tomentosa* extracts and purified components have shown a correlation between the anti-inflammatory activity and pentacyclic oxindole alkaloid content, particularly mitraphylline [23,24]. We have recently demonstrated that mitraphylline inhibits the nuclear factor (NF)-κB transcription factor and reduces the production of inflammatory cytokines tumor necrosis factor-alpha (TNF-α) and interleukin-6 (IL-6) in a model of LPS-stimulated mouse macrophages (RAW 264.7) [7].

Despite the clear relationship between the anti-inflammatory and antioxidant activities demonstrated in previous studies [25,26,27], the results presented here show that mitraphylline and isomitraphylline, when evaluated separately, did not reduce the accumulation of ROS in wild-type worms, and did not increase the survival rate of *C. elegans* exposed to a lethal dose of juglone (Figure 2B and Figure 3B). In contrast, ALE exhibited significant antioxidant activities in vivo, indicating that such effects may have resulted from a synergism between the secondary metabolites in the complex extract [28,29]. Indeed, a previous study demonstrated that the oxindole or pentacyclic alkaloid content were not responsible for the antioxidant activity of the *U. tomentosa* and *U. guianensis* extracts [30].

ALE enhanced resistance to oxidative stress in wild type worms, as shown by the reduced intracellular accumulations of ROS and the increased survival rates after a lethal dose of juglone. At 40 μg/mL, ALE switches from an antioxidant to a pro-oxidant behavior, increasing the ROS levels in a dose-dependent manner (Figure 2A). However, as the extract can only rescue the worms from a lethal dose of juglone at 80 μg/mL, we suggest that the slightly higher levels of ROS at this concentration induce the activation of the antioxidant defense system, effectively protecting the worms from acute oxidative stress (Figure 3A). Because of this interesting activity, we decided to further investigate the mechanism of action of ALE in the range of 40 to 100 μg/mL.

Pre-treatments with ALE led to the downregulation of genes associated with oxidative stress resistance, namely *sod-3*, *gst-4,* and *hsp-16.2* (Figure 4). These results indicate that ALE exhibits its antioxidant activity not only through the direct scavenging of ROS, but it also acts on the routes of oxidative stress detoxification, as SOD and GST act as antioxidant enzymes by converting ROS into less toxic compounds. SOD contributes to the first phase of detoxification, while GST acts in the second phase [31,32]. In addition to the catalytic activities of the *sod-3* and *gst-4* products, these genes also have a regulatory function activated by DAF-16 and SKN-1 transcription factors, respectively [33,34,35]. The small heat shock protein (HSP) 16.2 acts as a molecular chaperone, assuring protein stability in stress conditions.

In our study, pre-treatment with 80 μg/mL of ALE led to a decreased expression of *sod-3* in non-stressed worms, and had no effect in the worms subjected to juglone-induced oxidative stress (Figure 4A,B). Many studies on the antioxidant activity of plant extracts in *C. elegans* report an up-regulation of *sod-3* under basal stress conditions as proof of enhanced stress resistance and DAF-16 activation. At the same time, oxidative stress induced by environmental conditions, or artificially triggered by paraquat or juglone, has been shown by several groups to activate DAF-16 nuclear localization, and consequently induce the up-regulation of *sod-3*; however, the functional significance of this up-regulation is still not clear [36]. In a recent investigation using long-lived mitochondrial mutants, the authors show a clear association between the reduction of ROS levels mediated by antioxidants (e.g., ascorbic acid) and the down-regulation of the *sod-3* expression, leaving an open question for the possible roles of other transcription factors, such as SKN-1, in the regulation of *sod-3* expression [37]. With respect to these discordant and inconclusive interpretations, more studies are needed in order to fully understand the regulation of the *sod-3* expression by antioxidants in *C. elegans*.

As the expression of the *gst-4* and *hsp-16.2* in *C. elegans* is stress-inducible, both genes are often considered as markers for oxidative stress, and thus the ability of the antioxidant compounds to suppress their expression was reported in previous works as a sign of lower levels of oxidative stress. For example, Shi et al. [34] demonstrated that an ethanol extract of red mold rice enhanced resistance to oxidative stress in paraquat-stressed *C. elegans* worms, leading to a reduction of *sod-3* and *hsp-16.2* expression via the regulation of the DAF-16/FOXO pathway. Moreover, Kampkötter et al. [38] considered the reduction of the *gst-4* expression in worms under physiological conditions and under juglone-induced stress as a marker for the attenuation of oxidative stress conferred by a standardized extract from *Ginkgo biloba* leaves (EGb761). Taking these studies into account, the down-regulation of the stress-inducible genes *sod-3*, *gst-4*, and *hsp-16.2* exerted by ALE can be interpreted as a marker of reduced oxidative stress.

In order to investigate the involvement of DAF-16 and SKN-1 in the observed antioxidant effects, the mutants for the genes encoding these transcription factors were employed. ALE pre-treatment significantly reduced the ROS levels in the mutant worms lacking *daf-16* (CF1038), and also induced a slight, but not significant, increase in the survival rate of the same worms (Figure 5A and Figure 6A). In contrast, in the worms lacking *skn-1* (EU1), the ALE treatment had a minor, but significant, effect on the survival rate, and significantly increased the ROS accumulation (Figure 5B and Figure 6B). Taken together, these results indicate that SKN-1 is required for ALE-mediated stress resistance. In a previous study [39], an alkaloid-rich extract from roasted guarana seeds (*Paullinia cupana* var. *sorbilis*; Sapindaceae) with purine alkaloids was shown to enhance resistance against oxidative stress in *C. elegans*, and such an effect was attributed to the DAF-16 activation. These contradictory results suggest that the responses of *C. elegans* to extracts containing alkaloids are probably mediated by different stress signaling pathways. However, purine alkaloids and mitraphylline belong to different classes of alkaloids with different pharmacological properties.

To exclude caloric restriction as the promoter of the stress resistance induced by ALE, we studied its effect on *C. elegans* body size and on the growth of the bacteria used as a food source, *E. coli* OP50. In our study, the ALE treatment did not negatively interfere with both parameters, thus excluding this hypothesis.

## 4. Materials and Methods

The study was authorized by the Conselho Nacional de Desenvolvimento Científico e Tecnológico (CNPq), on behalf of the Conselho de Gestão do Patrimônio Genético/Ministério do Meio Ambiente (protocol no. 010102/2015–9).

### 4.1. Plant Material and Extracts

The leaves from *U. tomentosa* were collected at the Fazenda São João, located in Bannach, PA, Brazil (07°34′26′ S; 50°34′51′ W; altitude 415 m). The plant material was identified by Dr. Pietro Giuseppe Delprete (Herbier de Guyane, Institut de Recherche pour le Développement, Cayenne, French Guiana), and a voucher specimen was deposited into the Herbarium of Medicinal Plants at the Universidade de Ribeirão Preto (# HPMU-3133). The leaves were dried for 72 h at 45 °C in a circulating-air oven, reduced to powder, and passed through a 40-mesh sieve. A portion (20 g) of the powdered leaves was decocted with 1 L of distilled water for 30 min, filtered through filter paper, and subsequently freeze-dried to yield 3.2 g of crude ALE.

### 4.2. Analysis of Oxindole Alkaloids in the Leaf Extract

The oxindole alkaloids present in ALE were identified by UPLC-MS using a Waters (Milford, MA, USA) Acquity UPLC H-Class system equipped with a photodiode array detector and a Waters Xevo TQ-S tandem quadrupole mass spectrophotometer, with the electrospray source operated in the positive ion mode. Samples containing 1 mg/mL of ALE in methanol were passed through 0.45 μm pore size syringe filters, and aliquots (5 μL) were injected onto a Zorbax Eclipse (Agilent, Santa Clara, CA, USA) XDB-C18 column (150 × 4.6 mm i.d.; 3.5 μm particle size).

The mobile phase comprised ammonium acetate (0.2%) in water (solvent A) and acetonitrile (solvent B), supplied at a flow rate of 0.6 mL/min, with isocratic elution at 35% B between 0 and 18 min, linear gradient from 35% to 50% B between 18 and 32 min, isocratic at 50% B between 32 and 35 min, and a linear gradient from 50% to 35% B between 35 and 40 min. The source was maintained at 150 °C, the capillary voltage was 3.2 kV, the desolvation temperature was 350 °C, the flow rate of desolvation gas (N_2_) was 600 L/h, and the mass scan range was 100–600 *m*/*z* in the full-scan mode. The reference compounds, mitraphylline (CAS number 509-80-8; ≥95.0% purity) and isomitraphylline (CAS number 4963-01; ≥90.0% purity), were purchased from Sigma-Aldrich (St. Louis, MO, USA) and were used as standards.

### 4.3. Isolation of Isorhynchophylline from Crude Extracts

A crude extract prepared from the aerial parts of *U. tomentosa* (0.8 g) was solubilized in an ethanol–water solution (4:6), mixed with 500 mg of strong anionic resin (Dowex^®^ Marathon A, Sigma-Aldrich), and submitted to magnetic stirring for 30 min. An oxindole alkaloid-enriched fraction was obtained by washing the Dowex resin with ethanol–ammonium hydroxide (99.9:0.1) under magnetic stirring for 30 min, followed by rotary evaporation at 37 °C. A sample (10 mg) of the enriched fraction was submitted to semi-preparative high performance liquid chromatography (HPLC), using an Agilent Zorbax Eclipse XDB-C18 column (5 µ, 250 × 9.4 mm i.d.; 5 µ) with diode array detection (DAD) at 245 nm. The mobile phase comprised a 10 mM ammonium acetate buffer of pH 7.0 (solvent A) and acetonitrile (solvent B), supplied at a flow rate of 3 mL/min, with an isocratic elution at 35% B between 0 and 18 min, at 50% B between 18 and 32 min, and at 100% B between 32 and 40 min. The sample (3.0 mg) of pure isorhynchophylline as-obtained was dissolved in DMSO-d_6_ and submitted to 1D-nuclear magnetic resonance (NMR) spectroscopy (Appendix A) using a Bruker (Billerica, MA, USA) 600 MHz Avance III HD instrument coupled with a cryogenically cooled 5-mm dual probe optimized for ^13^C and ^1^H.

### 4.4. In Vitro Antioxidant Activity and Total Phenolic Content of U. tomentosa Leaf Extract

The DPPH assay was performed according to the method described by Blois [40], adapted for 96-well plates. Briefly, aliquots (100 µL) of ALE and 200 µM of DPPH (Sigma-Aldrich) were transferred to each well, and the microplate was maintained at room temperature (25 °C) in the dark for 30 min, following which the absorbance was measured at 517 nm. The scavenging activity was calculated according to Equation (1), as follows: (1)$$\mathrm{DPPH}\;\mathrm{scavenging}\;\mathrm{activity}\;\left(\%\right)=\left[\frac{A0-A1}{A0}\right]\times 100$$
where *A*0 is the absorbance of the reaction products in the presence of distilled water (control), and *A*1 is the absorbance of the reaction products in the presence of ALE. All of the assays were performed in triplicate, and the EC_50_ values were estimated by sigmoid non-linear regression and expressed as µg/mL.

The ABTS assay was performed according to the method described by Re et al. [41]. ABTS radical monocation was generated prior to use by mixing 7 mM of ABTS and 2.45 mM of potassium persulfate solutions, and leaving to rest at room temperature (25 °C) in the dark for 12 to 16 h. An ABTS radical cation working solution was obtained by diluting the reaction mixture with distilled water until the absorbance at 734 nm attained 0.70 ± 0.02. Aliquots of the ALE (100 µL) and ABTS working solution (250 µL) were transferred to each well and the microplate was maintained at room temperature (25 °C) in the dark for 7 min, following which absorbance was measured at 734 nm. A calibration curve was constructed using the water-soluble vitamin E analog Trolox (Sigma-Aldrich), with the concentration range of 0 to 0.040 mM as standard. All of the assays were performed in triplicate and repeated three times, and the results were expressed as µg of Trolox equivalents per mg of extract.

The FRAP assay, which is based on the ability to reduce the ferric tripyridyltriazine (TPTZ) complex to its ferrous form at a low pH, was performed according to the method of Benzie and Strain [42]. A FRAP working solution was prepared by mixing 300 mM of acetate buffer (pH 3.6), 10 mM TPTZ in 40 mM HCl, and 20 mM FeCl_3_.6H_2_O in proportions of 10:1:1 (by volume), and warming to 37 °C for 30 min immediately prior to use. Aliquots of the ALE (25 µL) and FRAP working solutions (175 µL) were transferred to each well, and the microplate was maintained at 37 °C in the dark for 7 min, following which the absorbance was measured at 595 nm. The spectrophotometric assay was calibrated using FeSO_4_.7H_2_O as the standard, and the FRAP values were expressed as µg of FeSO_4_ per mg of extract. All of the assays were performed in triplicate and repeated three times.

The Folin–Ciocalteu method was adapted to a 96-well microplate format. Briefly, 100 µL of the Folin–Ciocalteu reagent (Merck, Darmstadt, Germany) was added to 20 µL of sample; 5 min later, 80 µL of sodium carbonate (7.5% solution) was added. The reaction ran in the dark at room temperature for 2 h. The absorbance was measured at 750 nm using a microplate reader (Tecan Group Ltd., Männedorf, Switzerland). All of the measurements were carried out in triplicate and at least three times. The total phenol content was expressed as gallic acid equivalents (µg of GAE/mg of sample).

### 4.5. Maintenance of C. elegans Strains

The strains of *Caenorhabditis elegans* employed in the experiments were as follows: N2, wild type; CL2166, dvIs19 [(pAF15)gst-4p::GFP::NLS] III; CF1553, muIs84 [(pAD76) sod-3p::GFP + rol-6(su1006)]; TJ375, gpIs1 [hsp-16-2p::GFP]; CF1038, daf-16(mu86) I; and EU1, skn-1(zu67) IV/nT1 [unc-?(n754) let-?] (IV;V). *Escherichia coli* OP50 was provided to the worms as a food source. All of the organisms were obtained from the Caenorhabditis Genetic Center (CGC) at the University of Minnesota, Minneapolis, MN, USA.

The worms were cultivated on nematode growth media (NGM) plates that had been seeded with *E. coli* OP50 and incubated at 20 °C. In order to obtain age synchronized worms, gravid females were recovered from the plates and treated with a lysis solution (5 M NaOH, 5% NaOCl) for 5 min, so as to destroy the adult tissue while preserving the eggs. The lysate was pelleted by centrifugation (1 min, 1200 rpm) and the eggs were separated from the debris by density gradient centrifugation (4 min, 1200 rpm) with 5 mL of a 60% sucrose solution and 5 mL of sterile water. In order to remove the excess sucrose, the upper layer containing the eggs was transferred to a fresh tube, together with 5 mL of sterile water, and was centrifuged (1 min, 1200 rpm). The supernatant was discarded and the pelleted eggs were resuspended in a M9 buffer and allowed to hatch [43].

### 4.6. In Vivo Antioxidant Activity of ALE, Mitraphylline, and Isomitraphylline Using C. elegans as a Model Organism

#### 4.6.1. Intracellular Accumulation of ROS in Wild-Type Worms

The age synchronized N2 worms at L1 stage cultured in S-media inoculated with *E. coli* OP50 (OD = 1.0) were separated into groups and submitted to treatment with ALE at 10, 20, 40, 80, and 100 μg/mL; MIT at 1 and 2 μg/mL; and ISOMIT at 1 and 2 μg/mL. After 48 h at 20 °C, the worms were incubated at 20 °C for 1 h with 20 μM H2DCF-DA (FlukaChemie, Buchs, Switzerland), and were subsequently transferred to glass slides, paralyzed with a drop of 10 mM sodium azide, and observed under a Keyence (Neu-Isenburg, Germany) model BZ-9000 fluorescence microscope with λ_ex_ 488 nm and λ_em_ 540 nm. Live images were captured of at least 30 worms per treatment group, and the whole-body relative fluorescence was determined densitometrically using Image J software version 1.48 (National Institute of Health, Bethesda, MD, USA). The results were presented as mean fluorescence intensity ± standard error of the mean (SEM). The positive control comprised worms treated with 50 μg/mL of EGCG, while the untreated worms were employed as the negative control.

#### 4.6.2. Survival Rate of Wild-Type Worms Exposed to a Lethal Dose of Juglone

The age synchronized N2 worms at the L1 stage were separated into groups of 80 worms and were treated with ALE or oxindole alkaloids, as described in Section 4.6.1, for 48 h at 20 °C. Subsequently, all of the groups were exposed for 24 h to 80 μM of pro-oxidant juglone (5-hydroxy-1,4-naphthoquinone; Sigma- Aldrich), diluted in EtOH, and the number of dead worms were counted thereafter. The worms were considered dead when they did not respond to gentle touch with a platinum wire. The results were presented in terms of mean survival rate (%). The positive control comprised juglone-stressed worms were treated with 50 μg/mL of EGCG, while the juglone-treated worms were employed as the negative control.

### 4.7. Mechanism of the Antioxidant Activity of ALE in C. elegans

#### 4.7.1. Effect of ALE on the Expression Levels of the Oxidative Stress Resistance Related Genes: *sod-3*, *gst-4*, and *hsp-16.2*

The age synchronized worms at the L1 stage of the strains CF1553, CL2166, and TJ375 carrying a green fluorescent protein (GFP) gene were fused with *sod-3*, *gst-4*, and *hsp-16.2* promoter regions, respectively, and were employed to quantify the expression of the oxidative stress inducible genes in response to ALE. 

The worms of each strain were grown in an S-medium; separated into groups; and submitted to treatment for 72 h with ALE at 40, 80, and 100 μg/mL. To induce oxidative stress, a mild dose of 20 μM of juglone (prepared in EtOH) was added to the medium, and 24 h later, the worms were examined under a fluorescent microscope. The relative fluorescence of the posterior intestine (CF1553), whole body (CL2166), and head (TJ375) of the worms was determined densitometrically using Image J software. The results were presented as the mean relative fluorescence intensity ± SEM (vs. untreated worms or juglone-treated worms, as appropriate). The positive control comprised worms were treated with 50 μg/mL of EGCG, while the juglone-treated worms or untreated worms were employed as the negative controls, where appropriate.

#### 4.7.2. Involvement of DAF-16 and SKN-1 in the Antioxidant Activity of ALE: ROS and Survival Assays Using DAF-16 and SKN-1 Mutants

The age synchronized CF1038 and EU1 worms at the L1 stage were treated with ALE at 40, 80, and 100 μg/mL. The intracelullar levels of ROS and the survival rate after a lethal dose of juglone were evaluated as described above (Section 4.6).

### 4.8. Anti-Microbial Activity of ALE against E. coli OP50

The susceptibility of *E. coli* OP50 to ALE was performed using the well diffusion test recommended by the Clinical and Laboratory Standards Institute, employing a slightly modified version of the method described by Ashour et al. [44]. Briefly, bacteria from a single colony were grown on LB media at 37 °C for 24 h. Then, a cell suspension was prepared in a saline solution and adjusted to the 0.5 McFarland standard. The inoculum was spread evenly onto LB agar plates, and wells of 6 mm in diameter were punched out and loaded with 60 µL of ALE extract diluted in 100 µg/mL of sterile water.

### 4.9. Effect of ALE on C. elegans Body Size

The age synchronized N2 worms at the L1 stage cultured in S-media inoculated with *E. coli* OP50 (OD = 1.0) were separated into groups and submitted to treatment with ALE at 40, 80, and 100 μg/mL. After 48 h at 20 °C, the worms were transferred to glass slides, paralyzed with a drop of 10 mM sodium azide, and observed under a fluorescence microscope. Live images were captured of at least 30 worms per treatment group, and the body size was determined densitometrically using Image J software. The results were presented as mean area ± standard error of the mean (SEM).

### 4.10. Statistical Analyses

All of the analyses were performed using a GraphPad Prism for Windows software version 6.0 (GraphPad Software, La Jolla, CA, USA). The results were compared using one-way analysis of variance (ANOVA) followed by a Tukey’s post-hoc test.

## 5. Conclusions

In conclusion, we have demonstrated, for the first time, the ability of aqueous extracts, from *U. tomentosa,* to increase oxidative stress resistance in *C. elegans*. Our findings suggest that the antioxidant effects of the *U. tomentosa* leaf extract are independent of its alkaloid content, and that the mediated oxidative stress resistance requires SKN-1. This study sheds light on the potential therapeutic applications of *U. tomentosa* extracts in the prevention/treatment of diseases associated with oxidative stress, such as diabetes and other age-related diseases. Additionally, it proves that *C. elegans* is an appropriate model organism for investigating the modes of action of *U. tomentosa* extracts, as it provided new insights into the underlying mechanisms of action.

## Figures and Tables

**Figure 1 molecules-24-03299-f001:**
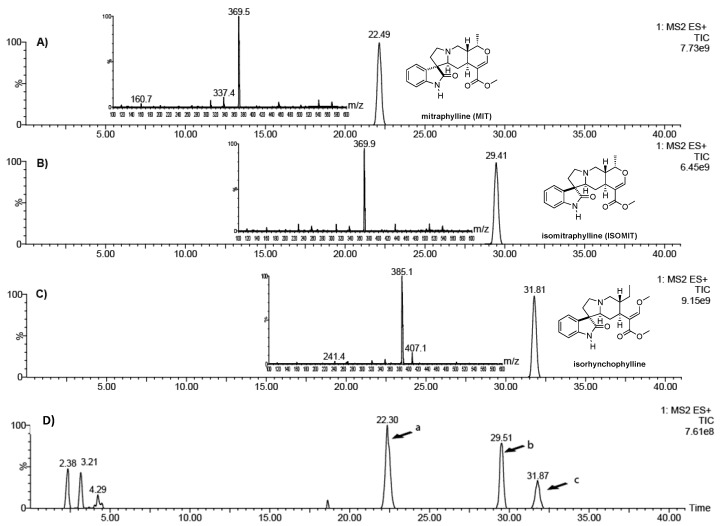
Ultra-performance liquid chromatography-mass spectrometry (UPLC-MS) chromatograms of (**A**) mitraphylline, (**B**) isomitraphylline, (**C**) isorhynchophylline, and (ALE) aqueous leaf extract from *Uncaria tomentosa* (**D**). The peaks are labeled (**a**) mitraphylline, (**b**) isomitraphylline, and (**c**) isorhynchophylline.

**Figure 2 molecules-24-03299-f002:**
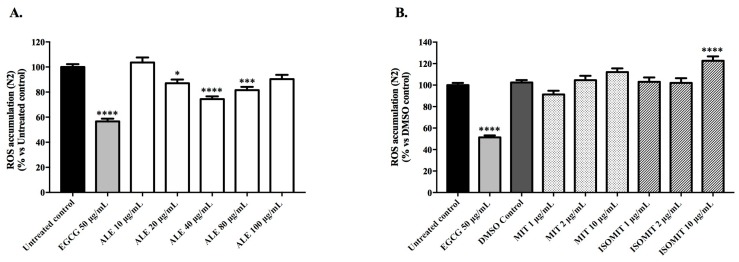
Intracellular reactive oxygen species (ROS) accumulation in wild-type N2 worms treated with *Uncaria tomentosa* leaf extract and its major compounds, isomitraphylline and mitraphylline, under physiological conditions, showing the following: (**A**) worms treated with ALE at 10–100 μg/mL; (**B**) worms treated with mitraphylline (MIT) and isomitraphylline (ISOMIT) at 1, 2, and 10 μg/mL. The control groups were untreated worms, pre-treated with dimethyl sulfoxide (DMSO), and pre-treated with epigallocatechin gallate (EGCG) at 50 μg/mL. Each bar represents the mean value ± standard error of the mean (SEM) from three independent assays. The asterisks indicate the statistical differences in relation to the untreated worms according to one-way analysis of variance (ANOVA), followed by a post-hoc Tukey’s test. * *p* < 0.05; ** *p* < 0.01, *** *p* < 0.001, and **** *p* < 0.0001.

**Figure 3 molecules-24-03299-f003:**
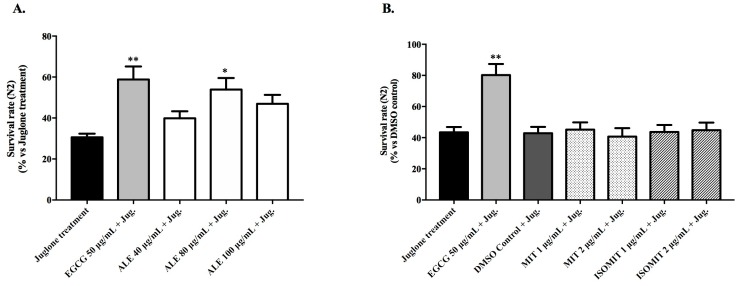
Survival rate of juglone-stressed (80 μM) wild-type N2 worms pre-treated with *Uncaria tomentosa* leaf extract and its major compounds isomitraphylline and mitraphylline, showing the following: (**A**) worms pre-treated with ALE at 10–100 μg/mL; (**B**) worms pre-treated with MIT and ISOMIT at 1 and 2 μg/mL. The control groups were the juglone-treated worms, pre-treated with DMSO, or juglone-stressed worms pre-treated with EGCG at 50 μg/mL. Each bar represents the mean value ± SEM from three independent assays. The asterisks indicate statistical differences in relation to the juglone-treated worms according to one-way ANOVA followed by a post-hoc Tukey’s test: * *p* < 0.05 and ** *p* < 0.01.

**Figure 4 molecules-24-03299-f004:**
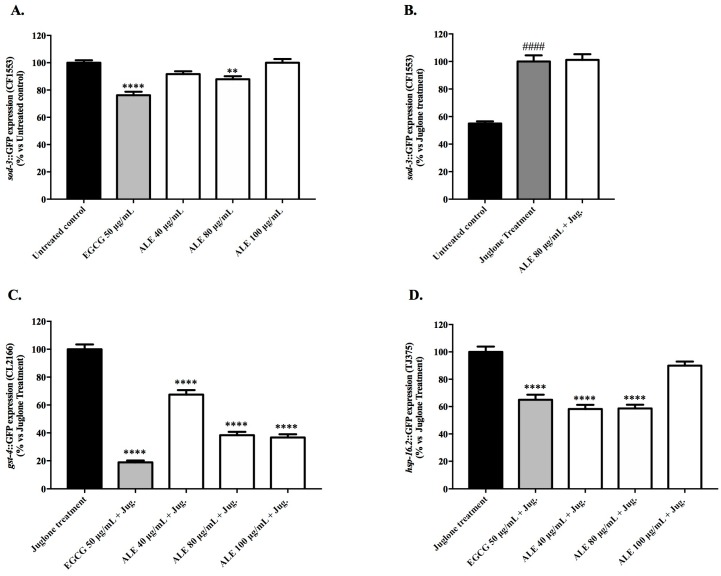
Effect of *Uncaria tomentosa* leaf extract on the expression levels of stress resistance related genes, showing the following: (**A**) *sod-3*::GFP expression in CF1553 mutant worms under basal stress conditions and (**B**) under juglone-induced oxidative stress; (**C**) *gst-4*::GFP expression in juglone-stressed CL2166 mutant strain and (**D**) *hsp16.2*::GFP expression in juglone-stressed TJ375 mutant worms treated with the ALE extract at 40–100 μg/mL.^.^ The control groups were the untreated worms, worms pre-treated with EGCG at 50 μg/mL, juglone-treated worms, and juglone-stressed worms pre-treated with EGCG at 50 μg/mL. Each bar represents the relative mean (*n* = 30) fluorescence intensity ± SEM from three independent assays. The asterisks indicate statistical differences (untreated worms vs juglone-treated worms) according to one-way ANOVA followed by a post-hoc Tukey’s test: ** *p* < 0.01 and **** *p* < 0.0001.

**Figure 5 molecules-24-03299-f005:**
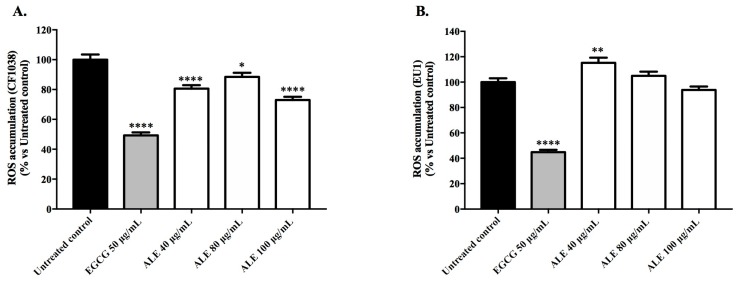
Intracellular ROS accumulation in the mutant worms treated with *Uncaria tomentosa* leaf extract under physiological conditions, showing the following: (**A**) mutant worms lacking *daf-16* (CF1038) and (**B**) mutant worms lacking *skn-*1 (EU1). The control groups were the untreated worms and those pre-treated with EGCG at 50 μg/mL. Each bar represents the mean value ± SEM from three independent assays. The asterisks indicate the statistical differences in relation to the untreated worms according to one-way ANOVA followed by a post-hoc Tukey’s test: * *p* < 0.05, ** *p* < 0.01, and **** *p* < 0.0001.

**Figure 6 molecules-24-03299-f006:**
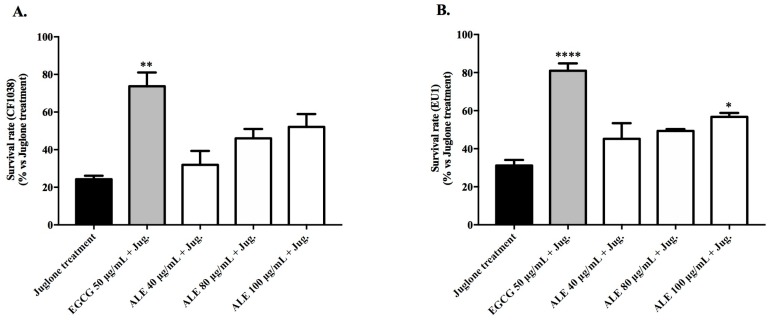
Survival rates of juglone-stressed (80 μM) *C. elegans* mutant strains pre-treated with 40–100 μg/mL of extract ALE from *Uncaria tomentosa*, showing the following: (**A**) mutant worms lacking *daf-16* (CF1038) and (**B**) mutant strains lacking *skn-*1 (EU1). The control groups were the juglone-treated worms or juglone-stressed worms pre-treated with EGCG at 50 μg/mL. Each bar represents the mean value ± SEM from three independent assays. The asterisks indicate the statistical differences in relation to the juglone-treated worms according to one-way ANOVA followed by a post-hoc Tukey’s test: * *p* < 0.05, ** *p* < 0.01, and **** *p* < 0.0001.

**Figure 7 molecules-24-03299-f007:**
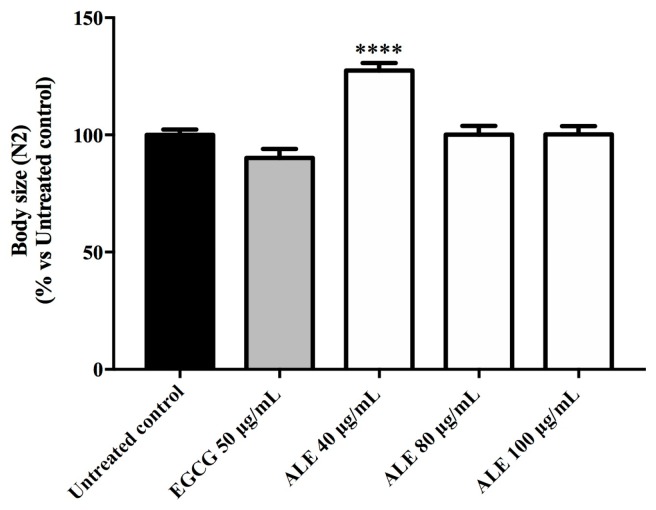
Effect of *Uncaria tomentosa* leaf extract on the body size of the N2 worms. The control groups were the untreated worms and the worms pre-treated with commercial green tea EGCG at 50 μg/mL. Each bar represents the relative mean (*n* = 30) area ± SEM from three independent assays. The asterisks indicate the statistical differences (vs untreated worms) according to one-way ANOVA followed by a post-hoc Tukey’s test: **** *p* < 0.0001.

**Table 1 molecules-24-03299-t001:** Antioxidant activity of *Uncaria tomentosa* leaf extract tested by in vitro assays.

Substance	ABTS mmol Trolox/mg AA and Extract	DPPH IC_50_ µg/mL	FRAP mM FeSO_4_/mg AA and Extract
Ascorbic acid (AA)	4.15 ± 0.15	2.05 ± 0.19	9.54 ± 0.54
ALE	2.15 ± 0.29	10.68 ± 0.64	2.9 ± 0.15

Abbreviations: ABTS—2,2′-azino-bis(3-ethylbenzothiazoline-6-sulphonic acid); ALE—aqueous leaf extract; DPPH—2,2-diphenyl-1-picrylhydrazyl; FRAP—ferric reducing antioxidant power.

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
