# Peer review of "Antioxidant Activity of an Aqueous Leaf Extract from Uncaria tomentosa and Its Major Alkaloids Mitraphylline and Isomitraphylline in Caenorhabditis elegans"

_molecules, 2019, doi:10.3390/molecules24183299_

Round 1

Reviewer 1 Report

It is opinion of the reviewer that this paper before acceptance in Molecules needs several corrections/modifications. My individual comments are listed below.

Title – It should be „Antioxidan Activity of an Aqueous Extract from Uncaria tomentosa Leaves and its Major Alkaloids”.

18 – It should be “The aim of this research was …”.

L  53 – What kind of damage?

75 – It should be “… in the extract (Fig. 1D).

Table 1 – Results of the ABTS assay should be reported as “mmol Trolox/mg extract” (mM= mmol/L).

Table 1 – This table must be changed because the results of ABTS and FRAP for ascorbic acid are reported per mg of AA not any extract.

100 – Extraction of EGCG from green tea should be described in material and methods section. 116 – Supplier of juglone? 116 – It should be “(80 μmol)”. 119 – The ration worms to EGCG solution should be reported.

Fig. 6 A & B – The differences between black and some white bars seems to be  significant. Please check results of Tukey’s test.

348 – It should be “ABTS radical cation working …”. 441 – It should be “… by Tukey’s post-hoc test”.

Elucidation of the chemical structure of three alkaloids by 1D and 2D NMR should be described/discussed in the Results section.

Author Response

Point 1: Title – It should be „Antioxidan Activity of an Aqueous Extract from Uncaria tomentosa Leaves and its Major Alkaloids”.

Response 1: We would like to thank the reviewer’s suggestion, however, since the expression “aqueous leaf extract” is broadly used and accepted, we decided to do not change the title of the manuscript.

Point 2: 18 – It should be “The aim of this research was …”.

Response 2: Changed in the manuscript.

Point 3: L  53 – What kind of damage?

Response 3: Changed in the manuscript.

Point 4: 75 – It should be “… in the extract (Fig. 1D).

Response 4: Changed in the manuscript.

Point 5: Table 1 – Results of the ABTS assay should be reported as “mmol Trolox/mg extract” (mM= mmol/L).

Response 5: Changed in the manuscript.

Point 6: Table 1 – This table must be changed because the results of ABTS and FRAP for ascorbic acid are reported per mg of AA not any extract.

Response 6: Changed in the manuscript.

Point 7: 100 – Extraction of EGCG from green tea should be described in material and methods section. 116 – Supplier of juglone? 116 – It should be “(80 μmol)”. 119 – The ration worms to EGCG solution should be reported.

Response 7: EGCG was commercially obtained. The supplier of juglone was added.

Point 8: Fig. 6 A & B – The differences between black and some white bars seems to be  significant. Please check results of Tukey’s test.

Response 8: It was no significance.

Point 9: 348 – It should be “ABTS radical cation working …”. 441 – It should be “… by Tukey’s post-hoc test”.

Response 9: Changed in the manuscript.

Point 10: Elucidation of the chemical structure of three alkaloids by 1D and 2D NMR should be described/discussed in the Results section.

Response 11: For this proposal, the compounds mitraphylline and isomitraphylline tested in this study were commercially obtained, thereby is not necessary to include NMR analysis. We have included the NMR 1D spectra of isorhynchophylline isolated from U. tomentosa in the supplementary information. Our research group have been isolated natural oxindole alkaloids from Uncaria tomentosa and U. guianensis species as mentioned recently in the literature (LOPES, A. A.; CHIOCA, B.; MUSQUIARI, B.; CREVELIN, E. J.; FRANÇA S. C.; DA SILVA, M. F. G. F.; PEREIRA, A. M. S. “Unnatural spirocyclic oxindole alkaloids biosynthesis in Uncaria guianensis, Scientific Reports, 9, 11349, 1-8, 2019).

Reviewer 2 Report

Before the manuscript will be accepted for publication, the authors must correct the manuscript by adding the following elements.
1. The authors have to add the conclusions because in this version the conclusions are included in discussion.
2. In the discussion is lack of mentioned in the introduction informations, except for anti-inflammatory action.
3. It was mentioned that the raw material contains polyphenols, so you can make an additional determination of the total content of polyphenolic compounds (Follin's reaction).
4. I do not like the numerous links to publications in references, I think that they should be saved in the form of specific publications.

Author Response

Point 1: The authors have to add the conclusions because in this version the conclusions are included in discussion.

Response 1: Changed in the manuscript.

Point 2: In the discussion is lack of mentioned in the introduction informations, except for anti-inflammatory action.

Response 2:  The antioxidant activity was also widely discussed in the discussion.

Point 3: It was mentioned that the raw material contains polyphenols, so you can make an additional determination of the total content of polyphenolic compounds (Follin's reaction).

Response 3: The total phenolic content was determined using Follin’s reaction and the results and methods were added to the manuscript. Eduardo J.Crevelin did the assay.

Point 4: I do not like the numerous links to publications in references, I think that they should be saved in the form of specific publications.

Response 4: The links were required by the journal.

Round 2

Reviewer 1 Report

The authors corrected this paper properly taken under considerations all my comments. Therefore, I can accept it now.

Reviewer 2 Report

The authors corrected this manuscript according to the reviewers suggestions. In this version the manuscript should be accepted for publication.